# Effectiveness of Semi-Supervised Active Learning in Automated Wound Image Segmentation

**DOI:** 10.3390/ijms24010706

**Published:** 2022-12-31

**Authors:** Nico Curti, Yuri Merli, Corrado Zengarini, Enrico Giampieri, Alessandra Merlotti, Daniele Dall’Olio, Emanuela Marcelli, Tommaso Bianchi, Gastone Castellani

**Affiliations:** 1eDIMES Lab, Department of Experimental, Diagnostic and Specialty Medicine, University of Bologna, 40138 Bologna, Italy; 2Dermatology, IRCCS Sant’Orsola-Malpighi Hospital, 40138 Bologna, Italy; 3Department of Physics and Astronomy, University of Bologna, 40127 Bologna, Italy; 4Department of Experimental, Diagnostic and Specialty Medicine, University of Bologna, 40138 Bologna, Italy

**Keywords:** computer-aided diagnosis, deep learning, image analysis, wound healing, image segmentation

## Abstract

Appropriate wound management shortens the healing times and reduces the management costs, benefiting the patient in physical terms and potentially reducing the healthcare system’s economic burden. Among the instrumental measurement methods, the image analysis of a wound area is becoming one of the cornerstones of chronic ulcer management. Our study aim is to develop a solid AI method based on a convolutional neural network to segment the wounds efficiently to make the work of the physician more efficient, and subsequently, to lay the foundations for the further development of more in-depth analyses of ulcer characteristics. In this work, we introduce a fully automated model for identifying and segmenting wound areas which can completely automatize the clinical wound severity assessment starting from images acquired from smartphones. This method is based on an active semi-supervised learning training of a convolutional neural network model. In our work, we tested the robustness of our method against a wide range of natural images acquired in different light conditions and image expositions. We collected the images using an ad hoc developed app and saved them in a database which we then used for AI training. We then tested different CNN architectures to develop a balanced model, which we finally validated with a public dataset. We used a dataset of images acquired during clinical practice and built an annotated wound image dataset consisting of 1564 ulcer images from 474 patients. Only a small part of this large amount of data was manually annotated by experts (ground truth). A multi-step, active, semi-supervised training procedure was applied to improve the segmentation performances of the model. The developed training strategy mimics a continuous learning approach and provides a viable alternative for further medical applications. We tested the efficiency of our model against other public datasets, proving its robustness. The efficiency of the transfer learning showed that after less than 50 epochs, the model achieved a stable DSC that was greater than 0.95. The proposed active semi-supervised learning strategy could allow us to obtain an efficient segmentation method, thereby facilitating the work of the clinician by reducing their working times to achieve the measurements. Finally, the robustness of our pipeline confirms its possible usage in clinical practice as a reliable decision support system for clinicians.

## 1. Introduction

Wound healing is a complex process where many factors, whether they are physical, chemical, or biological, work in balance to allow the repair of damaged tissue. Evaluating the impairments that may affect these factors is fundamental to ensure the greatest chances of healing acute and chronic ulcers in clinical settings [1].

A holistic approach to the patient’s situation is recommended, considering social conditions such as the possibility of obtaining access to care, their age, the caregivers’ presence, and a complete clinical and instrumental examination, which make it possible to recognize any comorbidities or deficits limiting the healing ability. Once an all-inclusive assessment of the wound has been performed and the correct therapy has been set, methodical and instrumental-assisted continuous monitoring becomes essential to ascertain whether the healing process is proceeding correctly or not [2].

A clinical wound follow-up involves the observation of different features in the ulcer site and leads physicians to infer data that can be used to determine prognosis and correct treatment. These parameters include the recognition of the wound margins, the bottom, the amount of exudate, the peri-wound skin, and its color, and finally, the shape and size of it [3]. Of all of them, the dimensions are the most informative values that are able to provide a numerical and objective quantification of the wound status, allowing us to plot the wound healing trajectory to determine whether it is proceeding in a correct fashion [1]. It is well known that an incorrect model of a wound assessment can lead to prolongation in wound healing [4]. Moreover, besides the clinical implications, shorter healing times due to better wound management may reduce the costs. Ulcer treatments are complex, and chronic ulcers have specific structures that need to be attended to, are more time and money consuming [5], and are known to enormously impact healthcare systems’ economic burden [6]. As an example of the health costs related to ulcers, regardless of the health model and the reference population, the economic price for their management is exceptionally high, with reported values of GBP 8.3 billion for the NHS in the UK in 2013 [7] and USD 31.7 billion for the Medicare system in the USA in 2018 [8]. These conservative calculations do not take comprehensive account of the private healthcare costs and cost–benefit and cost-effectiveness outcomes, but they provide an excellent example of the numbers accounting for wound care management. Additionally, we emphasize that financial burdens are growing in most Western countries [5,9]. Therefore, it is vital to try to the costs by pursuing the best and up-to-date aids, having well-trained specialists, and using reproducible methods for ulcer assessments [10].

Given the high inter- and intra-operator variability in collecting the physical characteristics [11,12], various hardware and software models have been tested to reduce operator fluctuations in chronic wounds measurements. Based on photographic support, some of these achieved the reasonably precise sizing of the ulcer, with results that are often superior to those of manual methods [12]. Moreover, manual measurement is time-consuming. Knowing that the video-assisted estimation of wound area is becoming one of the cornerstones of chronic ulcer management [13], we wanted to develop an artificial intelligence model based on wound pictures to provide accurate and automatically reproducible measurements.

The automatic segmentation of wounds is becoming an increasingly investigated field by researchers. Related studies on digital wound measurement mainly involve using video-assistive software or the use of artificial intelligence. The first ones used software coded to recognize specific image characteristics of the wounds (differences in color and saturation, grid scales, RGB tones, etc.) and provide a numeric value. On the other hand, the latter ones offer the possibility of obtaining measurements using different models, such as those ranging from machine learning supervised by humans and based on various classification methods (Naïve-Bayes, logistic regression, etc.) to unsupervised black-box-type models.

While classical software video-assisted classification methods provide a sufficient overall accuracy [14], AI sensitivity and specificity are promising, ranging from the 81.8% accuracy of classical neural networks up to the 90% accuracy of models such as the Generative Adversarial Network (GAN) [15]. The training of an artificial neural network usually requires supervision, but it is generally difficult to obtain great quantities of manually annotated data in clinical settings. Namely, the manual segmentation of images could be indeed extremely time consuming for large datasets. Though, when it is available, manual segmentation by human experts could further suffer from imperfections which are mainly caused by inter-observer variability due to a subjective wound boundary estimation [11]. Several approaches are apt to overcome the problem of large data annotations and the consequent image segmentation [16,17,18]. In this work, we proposed a combination of active learning and semi-supervised learning training strategies for deep learning models, proving its effectiveness for annotating large image datasets with minimal effort from clinicians. Moreover, to the authors’ knowledge, the resulting dataset, which is named *Deepskin*, represents one of the largest sets of chronic wounds used for the training of deep learning models. Therefore, the *Deepskin* dataset constitutes a novel starting point for deep learning applications in the analysis of chronic wounds and a robust benchmark for future quantitative results on this topic. Our study rationale is to develop an AI method, based on a convolutional neural network (CNN), enabling highly reproducible results thanks to its data-driven, highly representative, hierarchical image features [19]. Our future aims are to use the developed AI as a starting point to conduct more in-depth analyses of ulcer characteristics, such as the margins, wound bed, and exudation, which are used in the Bates-Jensen wound assessment tool (BWAT) score [20].

In this work, we started with a core set of 145 (less than 10% of the available samples) manually annotated images (see Section 4 for the details about the analyzed dataset). We implemented a U-Net CNN model for the automated segmentation task using an architecture that was pre-trained on the ImageNet dataset as a starting point. We repeated the active evaluation of automated segmentations until the number of training data reached (at least) 80% of the available samples. For each round of training, we divided the available image masks samples into training test sets (90–10%): in this way, we could quantify the generalization capacity of the model at each round (see Section 4 for the details about the implemented training strategy). For each round, we quantified the percentage of images that were correctly segmented on the validation according to the clinicians’ evaluation. To further test the robustness of the proposed training strategy and the developed *Deepskin* dataset, we compared the results obtained by our deep learning model also on public dataset [21,22], analyzing the generalization capability of the models using different training sets (Section 2 and Appendix A Appendix A).

## 2. Results

### 2.1. Training with Active Learning

To reach the target of segmenting 80% of the images, we conducted four rounds of training (including the first one) with our U-Net model, incrementing the number of training images/masks at each round. We started with a set of 145 images, validating the model using the remaining samples. The training rounds that were performed with the related number of samples used at each round are reported in Table 1. In each round, the two expert dermatologists evaluated the generated masks, noting the number of correctly segmented images that would be used in the next round of the training (Table 1).

The model was trained with the same set of hyper-parameters, resetting the initial weights in each round. We trained the model for 150 epochs, monitoring the scores described in Section 4.

### 2.2. Results on Deepskin Dataset

The results obtained by our training procedure at the end of the fourth round of training are shown in Figure 1a. We dichotomized the masks produced by our model according to a threshold of 0.5 (127 on the gray-level scale). The results shown in Figure 1b highlight the efficiency of the model in the wound detection and contours segmentation.

The results also prove the efficiency of the transfer learning procedure which was guaranteed by the EfficientNetb-3 backbone used in the U-Net model: after less than 50 epochs, the model achieved a stable DSC that was greater than 0.95. A transfer learning procedure was also proposed by Wang et al., but the model used in their work required more than 1000 training epochs, drastically increasing the computational time of the model training.

### 2.3. Results on Public Dataset

We tested our trained model on the FUSC public dataset without model re-training. In this way, we aimed to monitor the generalization capability of our model and its robustness against different images. The results obtained on these datasets are reported in Table 2.

The same dataset was also used by Wang et al. for training a MobileNetV2 model. We re-trained the same model (which is public available in [22]) on the FUSC dataset to achieve the reproducibility of the results. Furthermore, we tested the generalization capability of the Wang et al. model on our *Deepskin* dataset, allowing a direct comparison to be made between the two models and the robustness of the datasets used for their training. We report in Table 2 the results achieved by the two models on both datasets (*Deepskin* and FUSC), which are expressed in terms of the same metrics used above.

## 3. Discussion

The active learning strategy implemented in this work for the semi-automated annotation of a large set of images produced remarkable results. Starting with a relatively small core set of manually annotated samples, in only four rounds of training, we were able to obtain annotations for more than 90% of the available images. This procedure could drastically improve the availability of huge, annotated datasets with minimum timeframes and costs as required by expert clinicians.

We remark that the “classical” active learning involves a continuous interaction between the learning algorithm and the user. During training, the user is queried to manually label new data samples according to statistical considerations of the results proposed by the model. In our framework, we re-interpreted this procedure, requiring only a binary evaluation of the results by the user: if the generated segmentation did not satisfy the pre-determined set of criteria, it was excluded from the next training set, and vice versa. In this way, we can optimize the expert clinicians’ effort, minimizing the manual annotation requirement.

The *Deepskin* dataset introduced in this work constitutes one of the largest benchmarks available in the literature for the wound segmentation task. The heterogeneity of the wound images, combined with their high resolution and image quality, guarantees the robustness of the models trained on them. All of the images were acquired using the same smartphone photo camera, providing a robust standardization of the dataset in terms of image resolution and putative artifacts. At the same time, this characteristic poses the main limitation of this dataset, since batch effects could arise in terms of the generalization capability of the model. A robust data augmentation and deep learning model is crucial to overcome this kind of issue. This limit is a direct consequence of the single-center nature of our work, which reduce the possible heterogeneity of the images, i.e., the different conditions in picture acquisition and the operators available.

Furthermore, we would stress that all of the *Deepskin* images were acquired for clinical purposes. Therefore, the focus of each image is the estimation of the clinical wound scores using the photo. In this way, the amount of background information was limited as much as possible, and the center of each image was occupied by the wound area. This was not true when we used other public datasets, such as the FUSC one used in this work, for which a pre-processing was necessary [20].

Another limitation of the *Deepskin* dataset arises from the geographical location of the Dermatology Unit which collected the data. The IRCCS Sant’Orsola-Malpighi Hospital of the University of Bologna is an Italian hospital with a strong prevalence of Caucasian people. Most of the wound segmentation studies involve US hospitals with a greater heterogeneity of ethnicities. An artificial model trained on a dataset including an unbalancing number of Caucasian samples could introduce ethnicity artifacts and biases. In conclusion, a model trained on the only *Deepskin* dataset must take care of this limitation before any practical usage.

A third limitation of the results that we reported using the *Deepskin* dataset is related to the intrinsic definition of the wound area and the boundary. For both the initial manual annotation and consequent validation of the generated masks, we forced the model to learn that the wound area is the portion of the image that includes the core of the wound, excluding the peri-wound areas. Since there is not a standardized set of criteria for the wound area definition, its specification is made according to the clinical needs.

All of the above points must be considered when one is analyzing the results obtained by our model of the FUSC public dataset. The FUSC dataset includes low-quality images, without an evident focus on the wound area and with annotations based on different criteria. Furthermore, the dataset includes only foot ulcer wounds, which were acquired by US hospitals, with heterogeneous patient ethnicity. Nevertheless, the results showed in Table 2 confirm the robustness of our *Deepskin* dataset, as much as the robustness of our U-Net model which was trained on it. An equivalent result was obtained by comparing our U-Net model to the benchmark MobileNetV2.

Despite the unfair comparison of our deeper U-Net model with the lighter MobileNetV2 one, indeed, the generalization capability obtained by our architecture confirms the complexity of the wound segmentation task and the need for a more sophisticated architecture to address it.

## 4. Materials and Methods

### 4.1. Patient Selection

The analyzed images were obtained during routine dermatological examinations in the Dermatology Unit at IRCCS Sant’Orsola-Malpighi University Hospital of Bologna. The images were retrieved from subjects who gave their voluntary consent to the research. The study was approved by the Local Ethics Committee, and it was carried out in accordance with the Declaration of Helsinki. The data acquisition protocol was approved by the Local Ethics Committee (protocol n° 4342/2020 approved on 10 December 2020) according to the Helsinki Declaration.

We collected 474 patient records over 2 years (from March 2019 to September 2021) at the center, with a total of 1564 wound images (*Deepskin* dataset). A Smartphone digital camera (Dual Sony IMX 286 12MP sensors with a 1.25 µm pixel size, 27 mm equivalent focal length, F2.2 aperture, laser-assisted AF, and DNG raw capture) acquired the raw images under uncontrolled illumination conditions, various backgrounds, and image expositions for clinical usage. The selected patients represent a heterogeneous population, and thus, the dataset includes samples with ulcers at different healing stages and in different anatomical positions.

A global description of the dataset is shown in Table 3.

### 4.2. Data Acquisition

Two trained clinicians took the photos using a smartphone digital camera during clinical practice. No rigid or standardized protocol was used during the image acquisition. For this reason, we can classify the entire set of data as natural images with uncontrolled illumination, background, and exposition. The images were acquired in proximity to the anatomical region of interest, and the clinicians tried to put the entire wound area at the center of the picture. The photographs were taken without flash. The images were acquired according to the best judgment by each clinician, as it is standard in clinical procedure. For each visit, only one photo of each wound was collected. All of the images were captured in a raw format, i.e., RGB 8-bit, and saved in a JPEG format (1440 × 1080, 96 dpi, 24 bit).

### 4.3. Data Annotation

Two trained clinicians performed the manual annotation of a randomly chosen subset of images. The annotation was performed by one expert and reviewed by the second one to improve the data reliability. The manual annotation set includes a binary mask of the original image, in which only the wound area is highlighted; we intentionally did not try to define the peri-wound areas since it is not well confined, and thus, it is not representable with a binary mask. This set of image masks was used as the ground truth for our deep learning model.

Pixel-wise annotations are hard to achieve also for expert clinicians, and they are particularly time consuming. For this reason, we have chosen to minimize the number of manual annotations. This small core set of manual annotations was used as a kick starter for an active semi-supervised training procedure via a deep learning segmentation model. The initial set of segmentation masks was relatively rough, and it mostly consisting of polygonal shapes. This did not affect the following re-training procedure as it has already been observed that neural network models are able to generalize even from rough manual segmentation [23].

### 4.4. Training Strategy

Several machine learning and deep learning models have been proposed to address the automated wound segmentation problem in the literature [14,15,19,24]. Deep learning algorithms have provided the most promising results during this task. However, as for each segmentation task, the main issue is posed by the annotation availability. The annotation masks’ reliability and quality are as essential for the robustness of the deep learning models as their quantity is. The main drawback of deep learning models is the vast amount of training data required for the convergence of its parameters.

In this work, we propose a combination of active learning [16,25,26,27,28] and semi-supervised training procedure to address the problem of annotation availability, while minimizing the effort for clinicians. Starting with a core subset of manually annotated images, we trained a deep learning segmentation model on them, keeping the unlabeled images for the validation. Since no ground truth was provided for a quantitative evaluation of the model’s generalization capability, the segmentations generated by the model were submitted to the two expert clinicians. For each validation image, the clinicians determined if the generated segmentation was accurate according to the following binary criteria: (i) the mask must cover the entire wound area; (ii) the mask must cover only the wound area, i.e., the mask must not have holes or spurious parts; (iii) the mask shape must follow the correct wound boundaries. The validation images (and corresponding segmentation masks) which satisfied all of the criteria were inserted into the training set for a next round of model training. A schematic representation of the proposed training strategy is shown in Figure 2.

### 4.5. Segmentation Model

We tried several CNN architectures that are commonly used in segmentation tasks during the research exploration, starting with the lighter U-Net [29] variants and ending with the more complex PSPNet ones [30]. The evaluation of the model’s performances must balance having both a good performance on the validation set and a greater ability to extrapolate new possible samples. We would stress that while the above requirements are commonly looked for in any deep learning clinical application, they are essential in an active learning training strategy.

All of the predicted images were carefully evaluated by the experts of the Dermatological research group of the IRCCS Sant’Orsola-Malpighi University Hospital Dermatology Unit. Their agreement, jointly with the training numerical performances, led us to choose a U-Net-like [31] model as the best model that was able to balance our needs. In our architecture we used an EfficientNet-b3 model [32] for the encoder, adapting the decoder component accordingly. The evaluation of several encoder architectures during the preliminary phase of this work leads us to this choice, aiming to balance the complexity of the model and its performance metrics. Furthermore, the use of the EfficientNet-b3 encoder allowed to use a pre-trained model (on ImageNet [33] dataset) to kick-start the training phase.

We implemented the U-Net-like model using the Tensorflow Python library [34]. The model was trained for 150 epochs with an Adam optimizer (learning rate of 10^−5^) and a batch size of 8 images.

For each epoch we monitored the following metrics:$$precision=\frac{TP}{TP+FP}$$
$$recall=\frac{TP}{TP+FN}$$
$$DSC=\frac{2\times TP}{2\times TP+FN+FP}$$
where *TP*, *FP*, and *FN* are the True Positive, False Positives, and False Negative scores, respectively.

As a loss function, we used a combination of Dice score coefficient (DSC) and Binary Focal (BF) loss functions, i.e.,
$$DSC_{loss}\left(precision,\;recall\right)=1-\frac{\left(1+\beta^{2}\right)\left(precision\cdot recall\right)}{\beta^{2}\cdot precision+recall}$$
$$BF_{loss}\left(y_{true},\;y_{pred}\right)=-y_{true}\alpha {\left(1-y_{pred}\right)}^{\gamma }\log\left(y_{pred}\right)-\left(1-y_{true}\right)\alpha \;y_{pred}^{\gamma }\;log\left(1-y_{pred}\right)$$
$$Loss=DSC_{loss}+BF_{loss}$$
where $y_{true}\;$and $y_{pred}\;$are the ground truth binary mask and the predicted one, respectively. In our simulations, we used values of α = 0.25, β = 1, and γ = 2.

We performed an intensive data augmentation procedure to mimic the possible variabilities of the validation set. We provided possible vertical/horizontal flips, random rotation, and random shift with reflection for each image.

All of the simulations were performed using a 64-bit workstation machine (64 GB RAM memory and 1 CPU i9-9900K Intel^®^, with 8 cores, and a GeForce RTX 2070 SUPER NVIDIA GPU).

### 4.6. Testing on Public Dataset

Many photos are currently acquired during the clinical practice by several research laboratories and hospitals, but the availability of public annotated datasets is still limited. The number of samples required to train a deep learning model from the beginning is challenging to collect. A relevant contribution to this has been provided by the work of Wang et al. [21]. The authors proposed a novel framework for wound image segmentation based on a deep learning model, sharing a large dataset of annotated images, which had been collected over 2 years in collaboration with the Advancing the Zenith of Healthcare (AZH) Wound and Vascular Center of Milwaukee.

The dataset includes 1109 ulcer images that were taken from 889 patients and sampled at different intervals of time. The images were stored in PNG format as RGB (8-bit) and (eventually) zero-padded to a shape of 512 × 512. The same dataset was used also for the Foot Ulcer Segmentation Challenge (FUSC) at MICCAI 2021, and it constitutes a robust benchmark for our model.

The main difference between our dataset and the FUSC one is related to the heterogeneity of wound types. In our dataset, we collected wound images sampled in several anatomical regions (including feet), while the FUSC dataset is focused only on foot ulcer wounds. Moreover, also the image quality changes: the FUSC photos were sampled at more than 2× lower resolution with a different setup, but they were stored in a lossless format. For these reasons, the FUSC dataset represents a valid benchmark for the generalization capability of our model. We performed the evaluation of the entire FUSC dataset using our model, providing the same metrics that were used by Wang et al. for a direct comparison of the results.

## 5. Conclusions

In this work, we introduced a fully automated pipeline for the identification and segmentation of wounds in images that were acquired using a smartphone camera. We proved the efficiency of the training strategy discussed for the creation of the *Deepskin* dataset. We remark that with a minimal effort by the expert clinicians, the proposed active semi-supervised learning strategy allowed us to obtain an efficient segmentation method and a valid benchmark dataset at the same time.

We proved the segmentation efficiency of a CNN model trained on the *Deepskin* dataset, confirming the robustness of the dataset. The results obtained in this work confirms the possible usage of the proposed pipeline in a clinical practice as a viable decision support system for dermatologists. The proposed pipeline is currently being used in the Dermatology Unit of IRCCS Sant’Orsola-Malpighi University Hospital of Bologna in Italy, and we are currently working on overcoming the issues pointed out in this work. These improvements will be the subject of future works.

## Figures and Tables

**Figure 1 ijms-24-00706-f001:**
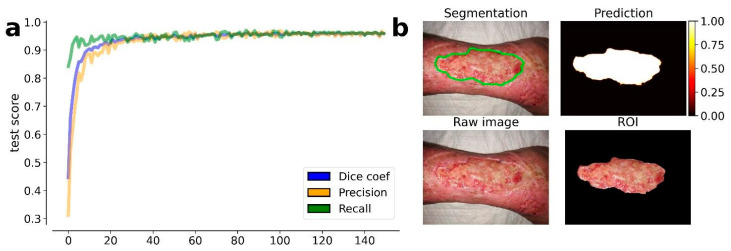
Results obtained by the trained U-Net model at the 4th round of training. (**a**) Evolution of the average metrics (dice coefficient, precision, and recall) during the training epochs (150). The metric values are estimated on the test set, i.e., the 10% of available images, which were excluded from the training set. On the top left image is the resulting segmentation. (**b**) On the top right image is the predicted segmentation mask. On the bottom left image is the raw (input) image. On the bottom right image is the resulting ROI of the wound area.

**Figure 2 ijms-24-00706-f002:**
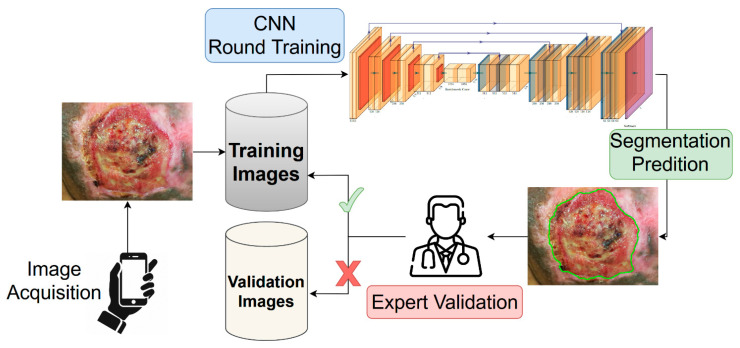
Representation of the active semi-supervised learning strategy implemented for the training of the wound segmentation model. The images acquired using a smartphone were stored into the training dataset. Starting with a small set of annotated images (not included into the scheme), we trained from scratch a neural network model for the wound segmentation. All of the unlabeled images were used as validation set, and the generated masks were provided by the expert. The expert analyzed the produced segmentation according to a predetermined evaluation criterion. The masks which satisfied the criteria would be added as ground truth for the next round of training.

**Table 1 ijms-24-00706-t001:** Results obtained by the U-Net model using the active semi-supervised learning procedure at each round. We report the number of images used for the training, the number of images used for the validation, the number of correctly segmented validation images (according to the expert evaluation), and the metric scores achieved (on the test set) after 150 epochs for each round of training, respectively. The percentages of training and validation images are referred to the whole set of available samples, i.e., 1564 images. The percentage of correct segmentation is referred to the total number of validated images per each round.

	Round 0	Round 1	Round 2	Round 3
N° training images	145 (9%)	368 (24%)	916 (59%)	1365 (87%)
N° validation images	1419 (91%)	1196 (76%)	648 (41%)	199 (13%)
N° correct segmentation	223 (16%)	548 (46%)	449 (69%)	112 (56%)
DSC metric	0.95	0.98	0.97	0.96
Precision metric	0.93	0.98	0.97	0.96
Recall metric	0.96	0.98	0.97	0.96

**Table 2 ijms-24-00706-t002:** Comparison of the results obtained by the MobileNetV2 model (Wang et al.) and the U-Net model proposed in this work, on the two available datasets, (a) *Deepskin* and (b) FUSC, respectively. We re-trained the MobileNetV2 on the FUSC dataset for the reproducibility of Wang et al. results. The U-Net model was trained only on the *Deepskin* dataset as described in Section 4.

a	Mobile NetV2	Our U-Net	b	Mobile NetV2	Our U-Net
DSC	0.64	0.96	DSC	0.90	0.78
Precision	0.53	0.96	Precision	0.91	0.83
Recall	0.85	0.96	Recall	0.90	0.72
	*Deepskin*			FUSC	

**Table 3 ijms-24-00706-t003:** (a) Description of the patient population involved in the study. We report the number of patients, which are split according to sex and age. (b) Description of the images involved in the study. We report the number of images split according to anatomical positions. The same wound could have been acquired at different time points. We report, in the last row, the number of images associated with each anatomical position.

*a*	Male	Female	Tot	b	Foot	Leg	Chest	Arm	Head	Tot
N° patients	210	264	474	N° wounds	97	354	14	6	2	473
Age	71 ± 17	77 ± 17	74 ± 20	N° images	364	1142	38	13	7	1564

## Data Availability

The data used during the current study are available from the corresponding author on reasonable request. The pre-trained model and parameters used for the image segmentation are available in the repository, *Deepskin* (https://github.com/Nico-Curti/Deepskin, accessed on 30 December 2022).

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
