# Peer review of "Effectiveness of Semi-Supervised Active Learning in Automated Wound Image Segmentation"

_ijms, 2022, doi:10.3390/ijms24010706_

Round 1

Reviewer 1 Report

In my opinion I think the authors have done a great job, it is scientifically consistent and with good results, congratulazioni. However, I would like to make some suggestions.

- At the beginning of the paper it is not entirely clear to me what the purpose of the study is. I understand that it is to make it easier to perform a data set, but that should be emphasized more, in my humble opinion.

- Along the same lines, if there is a lack of "what for", it seems to me that there is also a clear lack of "why". What is happening today that makes this contribution necessary?

- Another issue that comes to mind is the fact that CNN has already been used on numerous occasions for region selection in images. I think the authors should better highlight what is novel about the chosen CNN and the use with this type of imagery.

- One of the positive aspects of the paper is that they use photographs taken by physicians during their actual daily practice. However, this, which is something in favor, can be turned against when we see that only two physicians have taken these photos. I believe that using more physicians, each with smartphones of different makes, models, and ranges, in different settings, would have enriched the dataset.

- I see that the pictures taken have been validated by an expert (a human). Why are these photographs not directly validated by the person taking the photograph? What does this validation consist of? Validating one by one the photographs of the dataset can be tedious, what could be done to avoid this unit by unit control?

Author Response

In my opinion I think the authors have done a great job, it is scientifically consistent and with good results, congratulazioni. However, I would like to make some suggestions.

  • At the beginning of the paper, it is not entirely clear to me what the purpose of the study is. I understand that it is to make it easier to perform a data set, but that should be emphasized more, in my humble opinion.

We thank the reviewer for the suggestions. We have tried to better highlight the aims of our study in the Introduction section of the new version of the manuscript.

  • Along the same lines, if there is a lack of "what for", it seems to me that there is also a clear lack of "why". What is happening today that makes this contribution necessary?

We thank the reviewer for the suggestions. We have tried to better highlight the aims of our study in the Introduction section of the new version of the manuscript.

  • Another issue that comes to mind is the fact that CNN has already been used on numerous occasions for region selection in images. I think the authors should better highlight what is novel about the chosen CNN and the use with this type of imagery.

We apologize with the reviewer if the message conveyed by our work was about the introduction of a novel CNN architecture or improvement about existing ones. We tried to better emphasize the focus of our work in the new version of the manuscript (also according to the previous suggestions).

In our work we used a U-Net model as classical CNN model for segmentation task. The family of U-Net model involves all the auto-encoder CNN models (with the introduction of skip connections) independently by the format of the encoder part. In our work we used an EfficientNet-b3 backbone for the encoder part (and consequently adjusted the decoder one) as trade-off between computational efficiency and model complexity. Furthermore, the use of EfficientNet-b3 encoder allowed to use a pre-trained model (on ImageNet dataset) for the kick-starting of the training phase.

  • One of the positive aspects of the paper is that they use photographs taken by physicians during their actual daily practice. However, this, which is something in favor, can be turned against when we see that only two physicians have taken these photos. I believe that using more physicians, each with smartphones of different makes, models, and ranges, in different settings, would have enriched the dataset.

We agree with the reviewer that this kind of issues is certainly one of the main limits of our study and we have tried to better emphasize it in the conclusion sections. The limit pointed-out by the reviewer, indeed, is a direct consequence of the single-center nature of our study, which requires further analyses before the deployment of the model developed in large scale clinical applications.

  • I see that the pictures taken have been validated by an expert (a human). Why are these photographs not directly validated by the person taking the photograph? What does this validation consist of? Validating one by one the photographs of the dataset can be tedious, what could be done to avoid this unit-by-unit control?

The set of criteria used by the clinicians during the validation were described in the Training Strategy section of the main text (ref. lines 156 – 160).

As described in our work, the photos were acquired during clinical practice along 2 years. The clinicians involved in the acquisition are co-authors of our work and two of them performed the validation of segmentations. For time concerns we were not able to extend the validation to other expert clinicians. However, we surmised that the validation of two different experts could provide a reasonably good robustness of our results.

The classical training of a CNN requires the manual annotation (pixel-wise) of the whole set of available images. The time required for this task could be quantified in the demand of several minutes for each image by the clinicians. This amount of time is not comparable with minimum time required for a yes/no answer on the segmented image. The drastically reduction of time and effort by the clinicians appears, to authors knowledge, a sufficiently good improvement for the validation task.

Reviewer 2 Report

The article proposes a fully automated model based on an active semi-supervised learning training of a convolutional neural network model for the identification and segmentation of wound areas. Testeing the efficiency of our model against other public datasets proves its robustness. the results demonstrate the superiority of the proposed method. However, several issues should be addressed:

First, the main technical contributions of the article are not clear enough. More discussions should be provided in the Introduction section. 

Second, the literature review is not comprehensive. Active learning is widely used in medical image analysis, for example, a similar idea is used in Volumetric memory network for interactive medical image segmentation. In addition, recent semantic segmentation approaches should be included, like Rethinking Semantic Segmentation: A Prototype View. 

Third, I am not convinced with the experimental results. It is not reasonable or fair to compare U-Net with MoibleNet. What's the motivation for this comparison? If the authors indeed want to do such a comparison, more comprehensive experiments should be provided in terms of model size, FLOPs, rather than solely acc.

Author Response

The article proposes a fully automated model based on an active semi-supervised learning training of a convolutional neural network model for the identification and segmentation of wound areas. Testing the efficiency of our model against other public datasets proves its robustness. The results demonstrate the superiority of the proposed method. However, several issues should be addressed:

  • First, the main technical contributions of the article are not clear enough. More discussions should be provided in the Introduction section. 

We thank the reviewer for the suggestions, and we edited the Introduction section better highlighting the novelty of our work.

  • Second, the literature review is not comprehensive. Active learning is widely used in medical image analysis, for example, a similar idea is used in Volumetric memory network for interactive medical image segmentation. In addition, recent semantic segmentation approaches should be included, like Rethinking Semantic Segmentation: A Prototype View

We thank the reviewer for the suggested references. We added the references in the new version of the manuscript.

  • Third, I am not convinced with the experimental results. It is not reasonable or fair to compare U-Net with MobileNet. What's the motivation for this comparison? If the authors indeed want to do such a comparison, more comprehensive experiments should be provided in terms of model size, FLOPs, rather than solely acc.

We agree with the reviewer that the comparison between the two models (EfficientNet-b3 vs MobileNetV2) is unfair in terms of number of parameters and model complexity. However, in our work we want to use the results proposed by Wang et al. as benchmark for the robustness of the Deepskin dataset as training set of wound images. Therefore, the aim is to compare the two datasets and not the model performances directly. We edited the Introduction and Discussion sections of the new version of the manuscript to emphasize the reasons for this comparison, according to the reviewer suggestions.

Round 2

Reviewer 2 Report

The revision has addressed all my concerns.